

# Advances in Monitoring Black Sea Dynamics: A New Multidecadal High-Resolution Reanalysis

Leonardo Lima[1], Diana Azevedo[1], Mehmet Ilicak[2,1], Eric Jansen[1], Filipe Costa[1], Adil Sozer[1,3], Pietro Miraglio[1], Emanuela Clementi[1]

[1]CMCC Foundation – Euro-Mediterranean Center on Climate Change, Italy
[2]Eurasia Institute of Earth Sciences, Istanbul Technical University, Istanbul, Turkey
[3]Ordu University, Fatsa Faculty of Marine Science, Ordu, Turkey
*Correspondence to*: Leonardo Lima (leonardo.lima@cmcc.it)

**Abstract.** The Black Sea regional reanalysis serves as an essential tool for understanding the Black Sea's response to climate variability and advancing regional ocean monitoring efforts. In particular, the Black Sea reanalysis (BLK-REA) is built with high spatial resolution, 1/40° horizontal grid and incorporating 121 vertical levels. The model implementation includes lateral open boundary conditions (LOBC) at the Marmara Sea, allowing more accurate inflow/outflow dynamics through the Bosphorus Strait. BLK-REA assimilates sea level anomaly (SLA) and in-situ observations and applies a heat flux correction via sea surface temperature relaxation. Enhancements in data assimilation (DA) include an improved background error covariance matrix and an observation-based mean dynamic topography for SLA assimilation. When compared to available observations, the numerical results show high accuracy, with the largest temperature errors observed in the upper layers, primarily linked to the formation of the seasonal thermocline during the summer months. The SLA anomaly error is consistently around 0.02 m from the year 2000 onwards, and regions with elevated SLA errors are closely associated with the Rim Current and its mesoscale variability. Furthermore, BLK-REA plays a crucial role in generating Ocean Monitoring Indicators, which are essential for tracking and assessing the impacts of climate change in the Black Sea. For example, temperature data indicate ongoing warming in the 25 to 150 m layer, where the Cold Intermediate Layer is located. In addition, the Black Sea meridional overturning circulation has decreased from 0.1 Sv in 1993 to approximately 0.01 Sv in 2010, highlighting significant changes in the basin's circulation. The system is regularly updated, with the next version expected to improve both the model and DA components. For a future perspective, the next BLK-REA will expand the domain to include the Azov Sea and will feature an enhanced Bosphorus LOBC.

## 1 Introduction

The Black Sea is a semi-enclosed basin linked to the Marmara Sea through the Bosphorus Strait, the narrowest part of the Turkish Strait System (TSS). The TSS continues through the Dardanelles Strait, which connects the Marmara Sea to the Mediterranean. Salty waters originating from the Mediterranean Sea flow into the Black Sea through the TSS, serving as its main source of salinity. Despite this influx of saltier waters, the Black Sea is mainly considered a freshwater basin, characterized by a negative balance between evaporation ($E$), precipitation ($P$), and runoff ($R$): $E - P - R$. This imbalance is



compensated by a two-layer exchange through the Bosphorus Strait, where a stronger flow of the fresher upper layer moves southward toward the Marmara Sea (Beşiktepe et al., 1994; Altiok and Kayişoğlu, 2015). The surface circulation in the
Black Sea is primarily driven by the Rim Current, a semi-permanent cyclonic (counterclockwise) jet that flows along the edges of the basin. Along its path, this current interacts with multiple cyclonic gyres within its core and anticyclonic (clockwise) eddies along its peripheries, such as the Batumi and Sevastopol eddies (Oguz et al., 1993; Korotaev et al., 2003). An important feature of the Black Sea is the Cold Intermediate Layer (CIL), a cold water mass generated each winter through surface cooling and convective mixing. The CIL helps the ventilation of the sub-surface of the Black Sea (Özsoy
and Ünlüata, 1997). The CIL is typically defined by water temperatures below 8°C and extends between depths of 30 m to 80 m (Ivanov et al., 2001). The formation of the CIL in the Black Sea is primarily driven by convective processes during cold winters, where cool surface waters become denser and sink to intermediate depths. Recent studies have emphasized that CIL variability is not solely controlled by local winter conditions but is the result of a complex interplay between atmospheric forcing, lateral advection, and oceanic circulation (Korotaev et al., 2014; Miladinova et al., 2018; Capet et al.,
2020). For example, the Rim Current and associated mesoscale eddies create localized upwelling and downwelling regions, influencing the CIL distribution (Podymov et al., 2023).

Ocean reanalyses utilize state-of-the-art models that are constrained by atmospheric forcing and incorporate the best available observations through data assimilation techniques to reconstruct historical conditions. These products are crucial for monitoring, as they provide insight into the ocean's evolution in response to external forcing, and they allow for the
assessment of how environmental changes may affect marine biota, ecosystems, and activities dependent on the health of marine environments. The Black Sea reanalysis, developed within the framework of the Copernicus Marine Service, has been a valuable tool for enhancing our understanding of the Black Sea's response to climate change. For instance, its results demonstrated a recent surface warming of the Black Sea, identified through both sea surface temperature (SST) and subsurface temperature (Mulet et al., 2018; Lima et al., 2021). An ocean monitoring indicator (OMI) based on ocean heat
content (OHC) in the upper 300 m has also shown warming in the Black Sea. This increasing trend, as indicated by both reanalysis data and temperature measurements from Argo floats (Lima et al., 2020; Stanev et al., 2019), has contributed to the reduced presence of the CIL in the Black Sea in recent years.

Beyond OHC, Black Sea reanalysis has served as the foundation for other OMIs also produced within the scope of the Copernicus Marine Service. Using its velocity fields, Peneva et al. (2021) created an index for the Rim Current, showing that
the annual mean current speed fluctuated by approximately 30% between 1993 and 2019, with a positive trend of about 0.1 m s$^{-1}$ per decade. Ilicak et al. (2022) analyzed the meridional overturning circulation in the Black Sea and identified a strong correlation between the CIL and a newly proposed index representing the maximum overturning circulation in density space. Gunduz et al. (2021) proposed an index to characterize the upwelling system along the Turkish coast. Their study revealed significant year-to-year variations in upwelling intensity and duration, driven primarily by wind patterns. In addition, they
also found that recent declines in the CIL may have further influenced the properties of the upwelled waters.





Additionally, Black Sea reanalysis has played an important role in the practical development of the Black Sea physics forecast system (Ciliberti et al., 2022) and, more recently, in generating hourly datasets of velocity components and sea surface height for driving wave reanalysis within the framework of the Black Sea Monitoring and Forecasting Centre (BLK-MFC) under the Copernicus Marine Service (Ciliberti et al., 2021). One of the main challenges in developing a reanalysis for 70 the Black Sea is the scarcity of in-situ observational data to be assimilated, particularly in certain periods, such as the 1990s. This data scarcity is even more pronounced in deeper layers. The absence of observational data requires the use of a robust model capable of accurately simulating the physical processes involved in the Black Sea. Thus, significant progress has been made in improving the quality of the BLK-REA model component with respect to its previous version, including the implementation of a new configuration with lateral open boundary conditions (LOBCs) to better simulate exchange flows 75 through the Bosphorus Strait, such as the inflow of saltwater from the Marmara Sea. Also, the freshwater balance in the model has been refined by incorporating atmospheric forcing with hourly precipitation data, alongside monthly measurements of the Danube River discharge.

These improvements emphasize the importance of developing a regional reanalysis that integrates specific configurations and physical parameterizations tailored to accurately represent the unique characteristics of the Black Sea, which is 80 challenging to achieve with global reanalyses. The latter often rely on fixed parameter adjustments optimized for other regions in the global ocean. In addition, a regional reanalysis typically utilizes higher-resolution models, allowing for a more accurate representation of mesoscale and submesoscale processes, which are often unresolved or only partially captured by the coarse resolution of current global reanalyses.

This article is organized as follows: Section 2 provides a detailed description of the configuration for the new Black Sea 85 reanalysis (hereafter referred to as BLK-REA), which was released in December 2024. Section 3 discusses its main results. Finally, Section 4 summarizes the conclusions.

## 2 Methodology

Most of the methodology and configurations follow the previous Black Sea Reanalysis version (Lima et al., 2021), such that this section focuses on the main changes and enhancements present in the newly released version.

### 2.1 Ocean Model

The BLK-REA model component is the Nucleus for European Modelling of the Ocean (NEMO version 4.0, Madec and the Nemo team, 2019) configured for the domain (Azov and Marmara Seas are not included). NEMO is implemented at a horizontal resolution of 1/40° and 121 vertical geopotential levels. This horizontal resolution provides a spatial discretization of approximately 2.5 km, which conforms to the mesoscale eddy-resolving scale; the Rossby radius of deformation in the 95 Black Sea is approximately 20 km. The model is driven by atmospheric fluxes derived from ECMWF ERA5 reanalysis with spatial and temporal resolutions of 1/4° and 1 hour, respectively. The atmospheric forcing considers the following variables:



components of 10-m wind, total cloud cover, 2-m air temperature, 2-m dew point temperature, mean sea level pressure and precipitation. The system computes momentum, heat, and water fluxes at the air-sea interface using bulk formulae originally developed for the Mediterranean Sea (Castellari et al., 1998; Pettenuzzo et al., 2010), which have also been employed in
other Black Sea systems (Ciliberti et al., 2022; Lima et al., 2021). Additionally, the system applies daily sea surface temperature relaxation for heat flux corrections based on the ESA-CCI SST-L4 product (Good et al., 2020).

### 2.1.1 Lateral open boundary conditions

One of the key challenges in modeling the Black Sea dynamics is accurately simulating the outflow and inflow through the Bosphorus Strait. This is essential for correctly representing the surface and intermediate depth salinity patterns and sea
surface height (SSH) trends, as the Bosphorus acts as the sole passage for saltwater entering the Black Sea, and the only exit of the surface Black Sea water. The previous Black Sea reanalysis approach applied closed boundary conditions, requiring temperature and salinity restoration to achieve more accurate results. Additionally, SSH was controlled by treating the Bosphorus as an inverse river with a controlled flow to prevent artificial SSH trends. The present version incorporates open boundaries, using results from the Unstructured Turkish Straits System (U-TSS) model (Ilicak et al., 2021), leading to a
more accurate representation of these dynamics. U-TSS is built upon the Shallow Water Hydrodynamic Finite Element Model (SHYFEM; Micalleto et al., 2022). SHYFEM employs an unstructured finite element grid in the horizontal dimension and assumes hydrostatic approximation, solving depth-integrated shallow water equations in the vertical. The model features a horizontal resolution ranging from 500 meters in deeper regions to 50 meters in shallower areas, enabling a detailed representation of the Turkish Straits: Dardanelles and Bosphorus. Additionally, it incorporates 93 geopotential coordinate
levels in the vertical dimension. The current reanalysis simulation utilizes LOBCs from monthly-averaged fields of temperature, salinity, U and V velocity components, and SSH from a 4-year U-TSS simulation covering the period 2016–2019. Flather's boundary condition is applied to the barotropic component, while the flow relaxation scheme is utilized for tracers and baroclinic components, as implemented in NEMO. Custom interfaces between U-TSS and BLK-REA have been developed to adapt the U-TSS model outputs for the BDY module in NEMO (Chanut, 2005).

### 2.2 Observations

The system assimilates sea level anomaly (SLA) data from the dataset European Seas Along-Track L3 Sea Surface Heights Reprocessed, Tailored for Data Assimilation, available in the Copernicus Marine Service catalog (SEALEVEL_EUR_PHY_L3_MY_008_061, https://doi.org/10.48670/moi-00139; Faugère et al., 2022). To maximize the number and spatial coverage of in-situ observations assimilated into the model, we combine multiple datasets using a
predefined priority order, ensuring that duplicate profiles are excluded, as follows:

1. Global Ocean CORA In-situ Observations – Yearly Delivery in Delayed Mode from Copernicus Marine Service (INSITU_GLO_PHY_TS_DISCRETE_MY_013_001; https://doi.org/10.17882/46219) (Szekely et al., 2024).



2.  Global Ocean In-situ Near-Real-Time Observations from Copernicus Marine Service (INSITU_GLO_PHYBGCWAV_DISCRETE_MYNRT_013_030; https://doi.org/10.48670/moi-00036).

3.  SeaDataNet historical in-situ data collections (Myroshnychenko and Simoncelli, 2018; Myroshnychenko, 2020).

In data assimilation, the in-situ instrumental errors for temperature and salinity are depth-dependent, based on statistics from Ingleby and Huddleston (2007), while the in-situ representation errors are horizontally variable on the model grid, derived from previous model-observation statistics, and use same values for T and S. Both error components remain constant over time. For SLA observations, the instrumental error is fixed at 4 cm, while representation errors vary spatially and monthly, following Oke and Sakov (2008). Similar to the previous version, a background quality check is implemented in the data assimilation model to reject observations that deviate significantly from the model prior solution. An enhancement in the present version is the use of an observation-based mean dynamic topography (MDT) to compute the model-equivalent SLA in data assimilation. The MDT field is available in the Copernicus Marine Service catalog: https://doi.org/10.48670/moi-00138.

## 2.3 Data Assimilation

The data assimilation system, OceanVar, utilizes a three-dimensional variational (3D-Var) assimilation algorithm. OceanVar was initially developed for the Mediterranean Sea (Dobricic and Pinardi, 2008) and subsequently extended to the global ocean (Storto et al., 2011) and Black Sea (Ciliberti et al., 2022; Lima et al., 2021). The new system utilizes OceanVar, following the same equations outlined in Lima et al. (2021), with particular emphasis on the cost function ($J$) equation presented as follows:

$$J = \frac{1}{2}\delta x^T B^{-1}\delta x + \frac{1}{2}(H\delta x - d)^T R^{-1}(H\delta x - d) \tag{1}$$

where $\delta x = x_a - x_b$ is the increment, i.e., the difference between the analysis ($x_a$) and background ($x_b$), $d = y - H(x_b)$ is the misfit between an observation vector $y$ and its modeled correspondent (in the observation space) where $H$, the observation operator, maps the model fields to the observation locations. $B$ and $R$ are respectively the background and observation covariance matrices. $R$ is diagonal in the observation space and includes the sum of instrumental and representation errors, along with an additional error component that depends on the time difference between each observation and the analysis time. The latter component is weighted according to this temporal distance.

In OceanVar, the variational cost function is solved using the incremental formulation (Courtier, 1997), with preconditioning of the cost function minimization achieved through a change-of-variable transformation. Thus, to avoid inverting the $B$ matrix and to precondition the minimization of the cost function, the $B$ matrix is defined as $B = VV^T$ where $V$ is decomposed into a sequence of linear operators: $V = V_\eta V_h V_v$. The $V$ operator represents the background error covariance matrix, capturing the interdependencies among variables. Furthermore, a new control variable, $v = V^+ x$ (and thereby $x = Vv$), is introduced for the minimization process through the application of a transformation. Thus Eq (1) becomes:

$$J = \frac{1}{2}v^T v + \frac{1}{2}(HVv - d)^T R^{-1}(HVv - d) \tag{2}$$






In the present version, the linear operators $V_\eta$ and $V_h$ follow the same formulation described by Lima et al. (2021). Instead, $V_v$ incorporates background-error T and S vertical covariances that are modelled through 45-mode multivariate Empirical Orthogonal Functions (EOFs) and derives from a previous integration including the assimilation of SLA, T and S profiles, using the full model resolution. In addition, the new approach is non-stationary and a different set of EOFs are applied

considering the following decades: 1984–1993, 1994–2003, 2004–2013, and 2011–2020. EOFs are calculated for each month from anomalies estimated from daily T, S and SSH fields with respect to the long-term monthly mean of the corresponding decade.

## 2.4 Strategies and experiment setup

The experiment is initialized in 1991 with a rest state of temperature and salinity fields derived from the World Ocean Atlas

climatology (WOA 2018, Garcia et al., 2019). Following a spin-up of 2 years (1991-1992), the BLK-REA starts in 1993. The data assimilation is applied every 2 days, i.e., if the model initializes at time $t$, the next data assimilation cycle is performed at the time $t + 2$. The observation window is 4 days centered at the analysis time, so that each cycle assimilates observations from 2 days before until 2 days after the analysis time. In the Black Sea, the limited availability of in-situ observations for assimilation leads to systematic errors in certain variables during specific periods, particularly in the deeper

layers. To mitigate this bias, large-scale bias correction (LSBC) toward WOA2018 decadal climatologies is applied below 700 meters. The formulation and additional details on the LSBC scheme are described in Lima et al. (2021).

## 3 Results and discussion

This section presents quasi-independent validations of key variables from the BLK-REA, which consider both assimilated observations and those excluded during the data assimilation process due to specific adjustments (e.g., background quality

control). Additionally, it provides results for a set of OMIs computed from the BLK-REA.

### 3.1 Validation

Spatial seasonal maps of reanalysis SST are compared to satellite data in the period 1993-2022 and their difference shows a predominance of model negative biases in winter and spring, and positive biases in summer and fall, with a few exceptions as follows (Figure 1; left). Positive values of 0.1°C up to 0.3°C are visible in some regions during winter, such as close to the

Danube river mouth and along the southwest coast. Negative biases are present adjacent to the Kerch Strait, and the lowest negative biases, of more than 1.0°C, are exhibited in the upwelling region along the western Anatolian coast in summer. Most of the central-eastern area is covered by negative bias in fall. SST RMSD (Root Mean Squared Difference) maps indicate that errors are generally lower in spring and higher in fall (Figure 1; right). In general, larger values can be seen close to large river mouths such as near the Dnieper in winter and Danube in winter, spring, and summer. The highest errors,



exceeding 1°C, are observed along the western Anatolian Turkish coast in both summer and fall. In this region, a similar overestimation of upwelling phenomena was observed in the results of the previous Black Sea reanalysis, which was attributed to the influence of stronger winds (Lima et al., 2021). Recent analyses have indicated that the air-sea bulk formulation may be responsible for the intensified upwelling, and we plan to refine this model component in future releases of the Black Sea reanalysis.

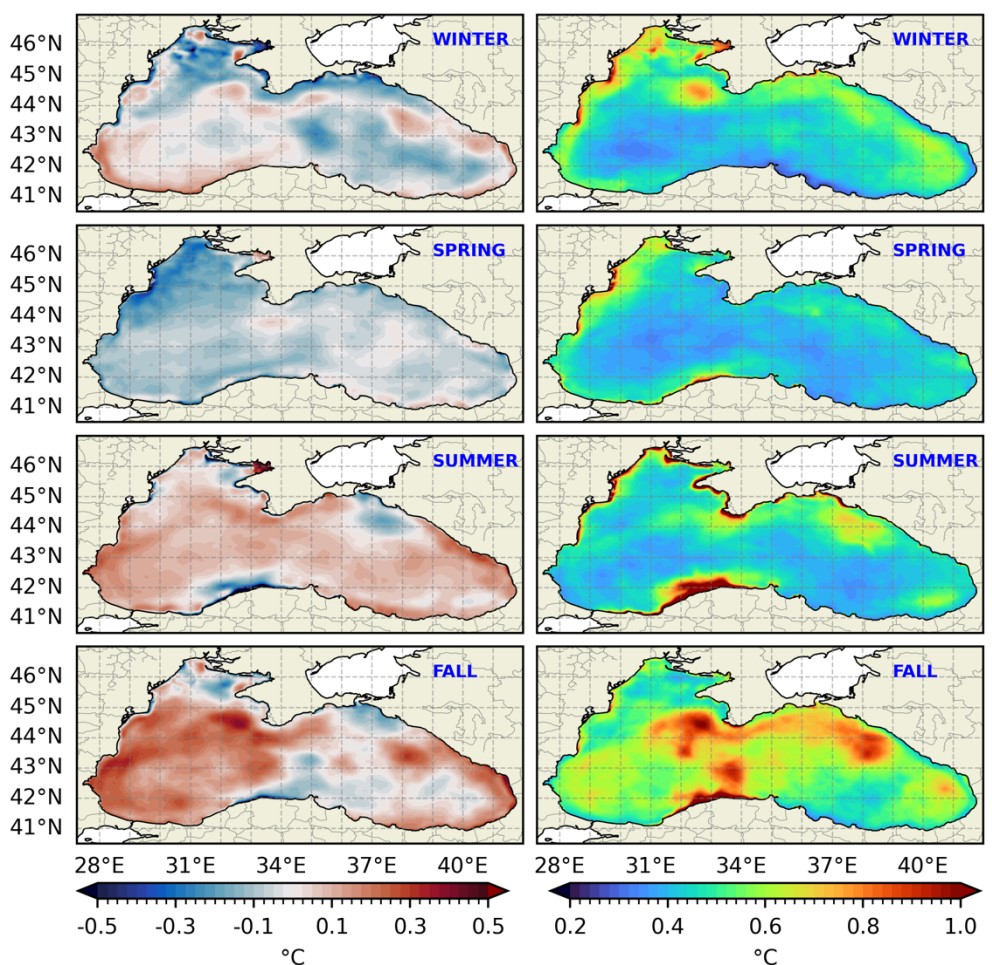

**Figure 1: Seasonal maps of the mean bias (left) and RMSD (right) of the SST (°C) with respect to the satellite ESA-CCI product over the period between 1993 and 2022. From top to bottom: winter, spring, summer, and fall.**

The Hovmöller diagram (time-depth) of RMSD for temperature computed as a lateral average, reveals a distinct seasonal cycle in the upper water column (Figure 2, top right), with lower errors in winter that increase during summer, exceeding 2°C in the upper 50 meters. This summer increase is associated with the model's misrepresentation of the seasonal thermocline, which is partially corrected through data assimilation. Below 100 m, errors remain low, staying below 0.25°C



for almost the whole period. Before the Argo floats era (mid-to-late 2000s), the scarcity of in-situ data limited the effectiveness of data assimilation, compromising both the model correction and the validation process.

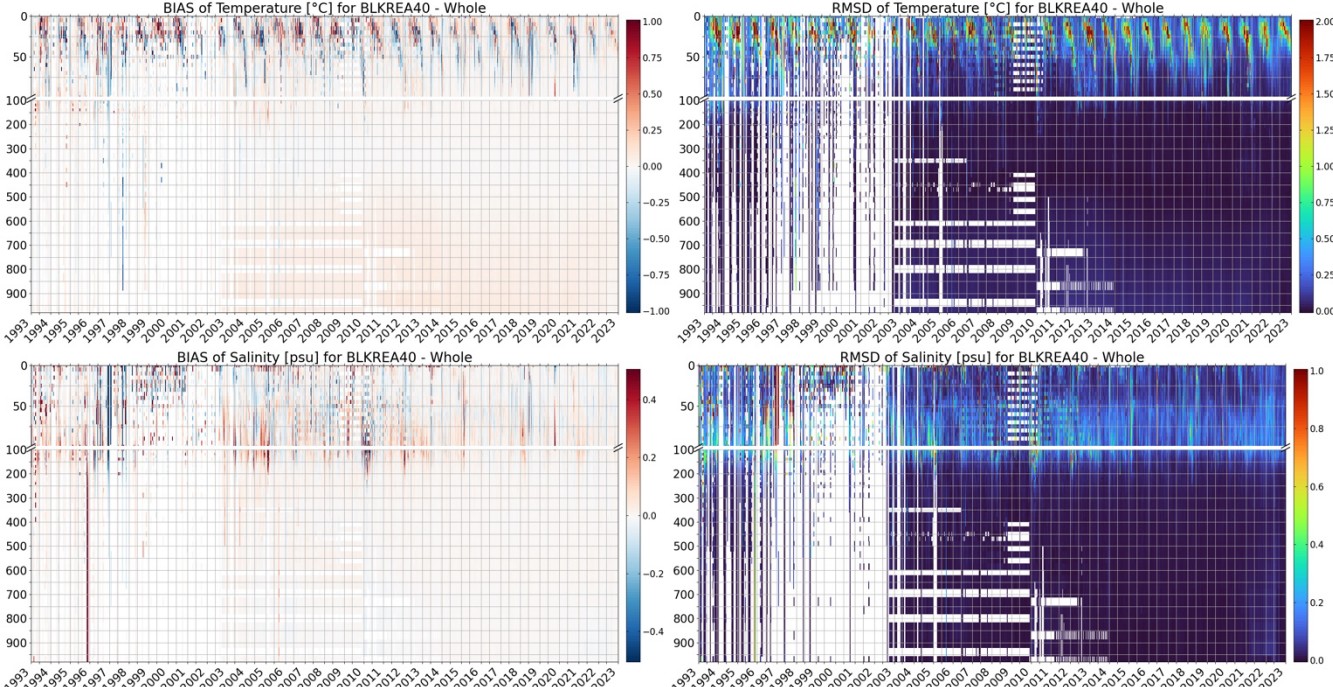


**Figure 2: Hovmöller (time-depth) diagrams of bias (left) and root mean square difference (right) computed against observations of temperature in °C (top) and salinity in PSU (bottom) available in the Black Sea domain from 1 January 1993 to 31 December 2022.**

The Hovmöller diagram of temperature biases predominantly shows positive values, occasionally exceeding 1°C near the

surface, with intermittent periods of negative biases (Figure 2, top left). The most pronounced discrepancies are observed within the seasonal thermocline depths. There is a tendency for biases to shift from positive in upper layers to negative values at deeper layers, down to 100 m. This may be related to the misrepresentation of the vertical position of the seasonal thermocline in the BLK-REA results compared to observations over time.

Unlike temperature, the Hovmöller diagram of RMSD for salinity does not exhibit a clear seasonal cycle (Figure 2, bottom

right). Errors exceed 1 psu during short periods, particularly near the surface. Apart from these peak values, errors are relatively higher between 50 and 100 m. Within this layer, errors tend to decrease over time, reaching values below 0.25 psu in the most recent years of validation. Once again, the scarcity of observations compromises the validation process before 2004. Below 200 m, errors remain very low, with values consistently below 0.1 psu.

Salinity biases show a predominance of positive values (Figure 2, bottom left). The Hovmöller diagram reveals two main

characteristics: BLK-REA diverges more from observations in the upper 200 m, with values alternating between positive and negative biases. Below 200 m, biases approach zero and remain predominantly positive. Since 2014, the values tend to be closer to zero, with biases showing a relative reduction in the upper 200 m.



Figure 3 presents the temporal and spatial averaged RMSD and biases for temperature, comparing the reanalysis results with in-situ observations. For a better spatial analysis, we divide the Black Sea into three different regions: western, central, and eastern. The largest temperature errors occur in the upper layers and are primarily associated with the formation of the seasonal thermocline in summer, as shown by the Hovmöller diagram (Figure 2, top). In this layer, RMSD reaches a maximum of approximately 1.5 °C in the western and eastern regions, and about 1.25 °C in the central region. In the upper water column, temperature biases are negative in the eastern and central regions. In the western region, the bias is positive from the surface down to 50 meters, reaching values of up to 0.1 °C, and becomes negative between 50 and 100 meters, aligning with the values observed in the central region. The largest errors are observed in the eastern region. Errors gradually decrease with depth, with RMSD reaching values lower than 0.25°C at 150 m, biases becoming closer to zero.

Figure 3: Vertical profiles of the RMSD (left panel), bias (middle panel) and number of observations (right panel) for temperature (in °C), by comparing the reanalysis results against in-situ profilers in three areas (West, Central and East) of the Black Sea domain from 1 January 1993 to 31 December 2022.



The same analysis is provided for salinity showing highest error and bias values in the upper 100 m in the western region, with an error exceeding 0.8 psu near the surface. RMSD values are lower in the central and eastern regions, where errors remain below 0.2 psu throughout the water column (Figure 4). Bias values are in general small, and in particular in the
western region, biases are high and positive near the surface, rapidly transitioning to negative values, reaching -0.04 psu between 25 and 50 meters. In contrast, salinity biases in the surface layers of the central and eastern regions are close to zero, but they increase in the subsurface starting at 50 meters, following the values observed in the western region. At a depth of around 100 meters, all regions exhibit relatively high salinity biases, with the eastern region showing the highest values, reaching up to 0.04 psu. In general, salinity biases decrease beneath 100 meters and approach zero at depths greater than 200
meters.





**Figure 4: Vertical profiles of the RMSD (left panel), bias (middle panel) and number of observations (right panel) for salinity (psu), by comparing the reanalysis results against in-situ profilers in three areas (West, Central and East) of the Black Sea domain from 1 January 1993 to 31 December 2022.**


The time series of spatially averaged SLA RMSD (Figure 5) indicates that, after the year 2000, the BLK-REA demonstrates strong performance with high skill and errors typically around 0.02 m. This is well within acceptable limits, as it remains below the observation error for SLA in data assimilation, which is 0.04 m. It is to be noted that the evaluation is performed in areas deeper than 1000 m, considered as the level of no motion when assimilating SLA data through a dynamic height

operator. Spatial maps of the sea level anomaly RMSD reveal the highest values, ranging from 2.5 to 4 cm, with occasional peaks exceeding 4 cm, predominantly along the basin's periphery (Figure 6). These elevated deviations are closely linked to the Rim Current and its inherent mesoscale variability. In contrast, SLA RMSD values in the central basin are generally lower, around 2 cm. Notably, areas with large RMSD values align with regions of strong eddy kinetic energy (EKE), which we use to assess mesoscale activity (Figure 7). This pattern is particularly evident along the Anatolian, Caucasian, and

Crimean coasts, where well-known mesoscale characteristics are present (Koroatev et al., 2003). SLA errors show slight variations across seasons, with particular attention given to high values along the Caucasian coast. These values extend further offshore in winter. Additionally, higher SLA errors occupy a larger area around the Batumi eddy region during fall. Error values are relatively high along the Crimean coast, particularly in the western region, where the persistent influence of the Sevastopol eddy contributes to elevated mesoscale variability. However, these errors tend to decrease during summer.

Kubryakov and Stanichny (2015) reported that the total number of detected eddies exhibits local maxima in both the Sevastopol and Batumi eddy regions, reflecting the dynamic nature of coastal circulation in these areas. These complex dynamics may not be adequately resolved by the model, potentially contributing to elevated SLA errors in these areas.



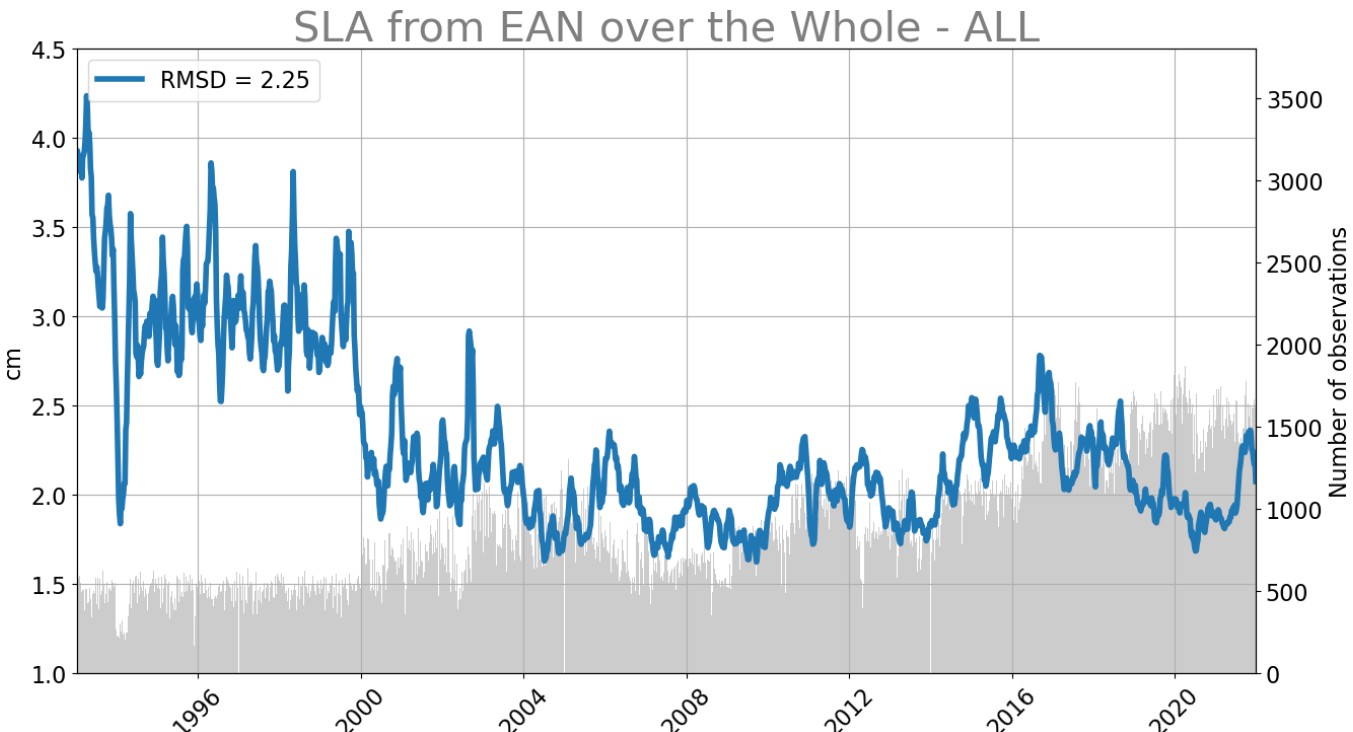

**Figure 5: Time evolution of basin-averaged SLA RMSD, comparing BLK-REA with satellite along-track SLA data, with a 7-day moving average applied to smooth short-term variability.**

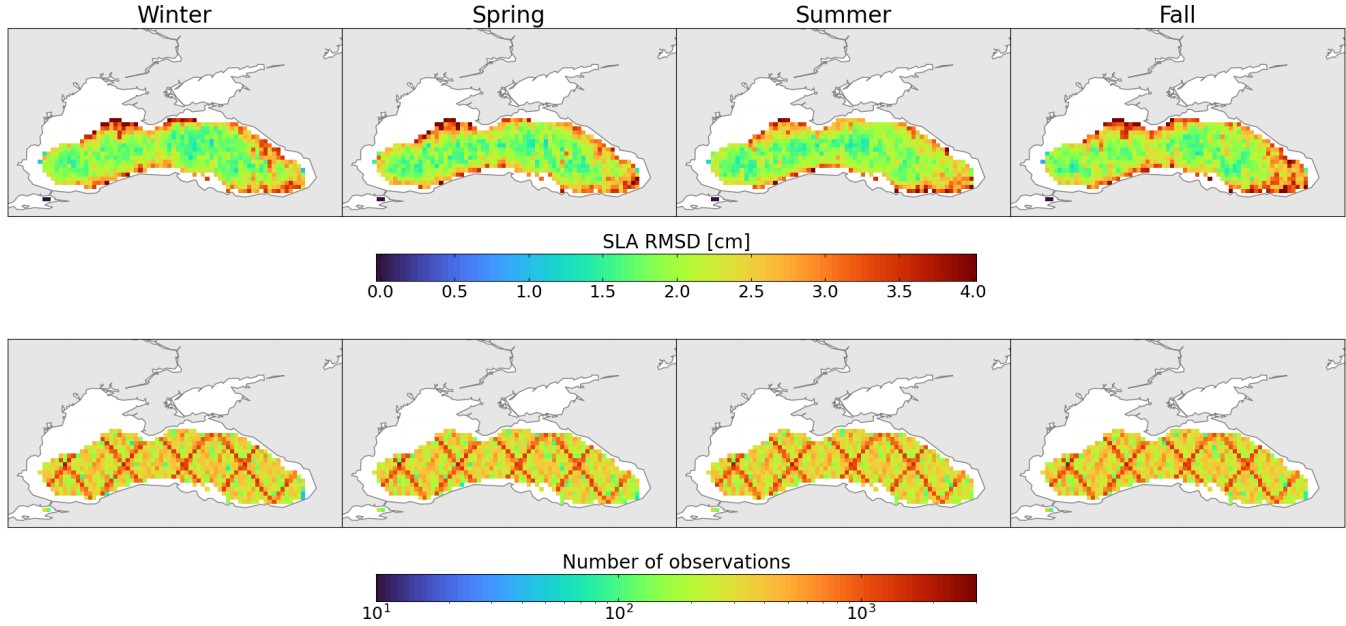

**Figure 6: Seasonal maps of the mean RMSD (top) of the SLA (cm) with respect to the satellite along-track SLA product over the period between 1993 and 2022. Bottom maps indicate the number of observations for each season.**



This interpretation is further supported by EKE maps (Figure 7), which consistently show high energy levels in these

regions, indicating intense mesoscale activity that aligns with the SLA discrepancies viewed in Figure 6. EKE maps show

elevated values along the Rim Current, particularly along the Caucasian coast. EKE values from the BLK-REA generally

exceed those derived from altimetry, likely due to the lower resolution of the Level 4 altimetric product, which has a spatial

resolution of 0.25 degrees. In fact, BLK-REA EKE results surpass the altimetry-derived estimates, showing peak values of

approximately 300 cm² s⁻² in the Batumi and Sevastopol eddies during fall and spring, respectively, while altimetry-based

EKE in the same regions and seasons reaches around 220 cm² s⁻².



**Figure 7: Seasonal maps of eddy kinetic energy (EKE) (cm² s⁻²) from satellite SLA L4-product and BLK-REA over the period between 1993 and 2022, based on the surface geostrophic velocity component.**






### 3.2 Ocean monitoring indicators

We present a set of OMIs computed from the BLK-REA: ocean heat content, Rim Current interannual variability, and meridional overturning circulation. These indicators provide valuable insights into key aspects of Black Sea dynamics.

### 3.2.1 Ocean heat content

Figure 8 illustrates the time evolution of basin-averaged temperature, with the 8 °C isotherm selected to track the Black Sea CIL over time. The formation of the CIL is mainly associated with water cooling during the winter season, and its presence is consistently observed until 2008. From 1993 to 2008, the CIL extends from the surface to approximately 100 meters in depth. After 2008, this pattern exhibits a significant shift, as temperatures rise, leading to the frequent disappearance of the CIL. Nevertheless, instances of CIL formation are also observed in 2012, with a reduced extent in 2017, consistent with

Argo float measurements (Stanev et al., 2019). More recently, a very weak CIL formed in March 2022, as documented by Çokacar et al. (2024), who attributed this event to intense cold-air masses that caused severe weather conditions across southern Europe, including the Black Sea, and influenced CIL formation.



**Figure 8: Hovmoller (time-depth) diagrams of monthly basin-averaged temperature in °C (top) and anomaly of temperature in °C**
**(bottom). The monthly anomaly estimates considered the climatological period 1993–2014 of each corresponding month. The blue dashed line indicates the mean position of the 8°C isotherm (top).**

The warming signal is evident in the Hovmöller diagram of temperature anomalies, which shows a predominance of positive values starting in 2009. Occasionally, positive values are interrupted by negative anomalies in the upper layers, as seen in years with the presence of the CIL: 2012, 2017, and 2022.

The analysis of ocean heat content in the Black Sea follows the formulation outlined by Lima et al. (2020), as described by the equation below:

$$OHC = \int_{z_1}^{z_2} \rho_0 c_p (T_m - T_{clim}) dz \tag{3}$$



with $\rho_0$ equal to 1020 kg m$^{-3}$ and $c_p$ equal to 3980 J kg$^{-1}$°C$^{-1}$ are, respectively, the density and specific heat capacity; and $dz$ indicates a certain ocean layer limited by the depths $z_1$ and $z_2$; $T_m$ corresponds to the monthly averaged temperature and

$T_{clim}$ is the climatological temperature of the corresponding month. In this study, OHC is calculated as the deviation from the reference period of 1993–2014.

The OHC in both the 0-10 m and 0-100 m layers shows an overall warming trend, with values of 0.08 W m$^{-2}$ and 0.59 W m$^{-2}$, respectively (Figure 9). Table 1 shows OHC trends within other layers to compare the values with those reported by Lima et al. (2021), who analyzed OHC trends using the previous Black Sea reanalysis (Lima et al., 2021) during the period

1993–2018. In general, the newest BLK-REA shows lower OHC trends. In the 0-10 m layer, the OHC curve shows several positive peaks around $1 \times 10^8$ J m$^{-2}$ in 2010, 2012, 2018, and 2020. In contrast, negative peaks are observed in 1993, 1997, and 2001. An interesting observation is that although the CIL is present in 2012, there are positive anomalies in the upper layers that year. This suggests that colder waters from the upper layers, which subducted in 2011, may have reached the CIL levels in 2012. These features are visible in the Hovmöller diagrams of basin-averaged temperature anomalies (Figure 8).

The CIL signal is clearly present in the 0-100 m layer in 2012. Additionally, the OHC shows a clear agreement with the CIL cold content observed in the data, which was estimated by Capet et al. (2020) using temperature observations from various platforms. Specifically, years of higher heat content correspond to a reduction in CIL cold content, while years of lower heat content coincide with an increase in CIL cold content. In more recent years, the CIL cold content values are nearly zero, except for 2012 and 2017, when the values exceeded $1.5 \times 10^8$ J m$^{-2}$ in 2012 and were slightly below this threshold in 2017;

see the green curve in Figure 9. Correspondingly, both years show a decrease in OHC, reinforcing the relationship between CIL cold content and heat content variability.



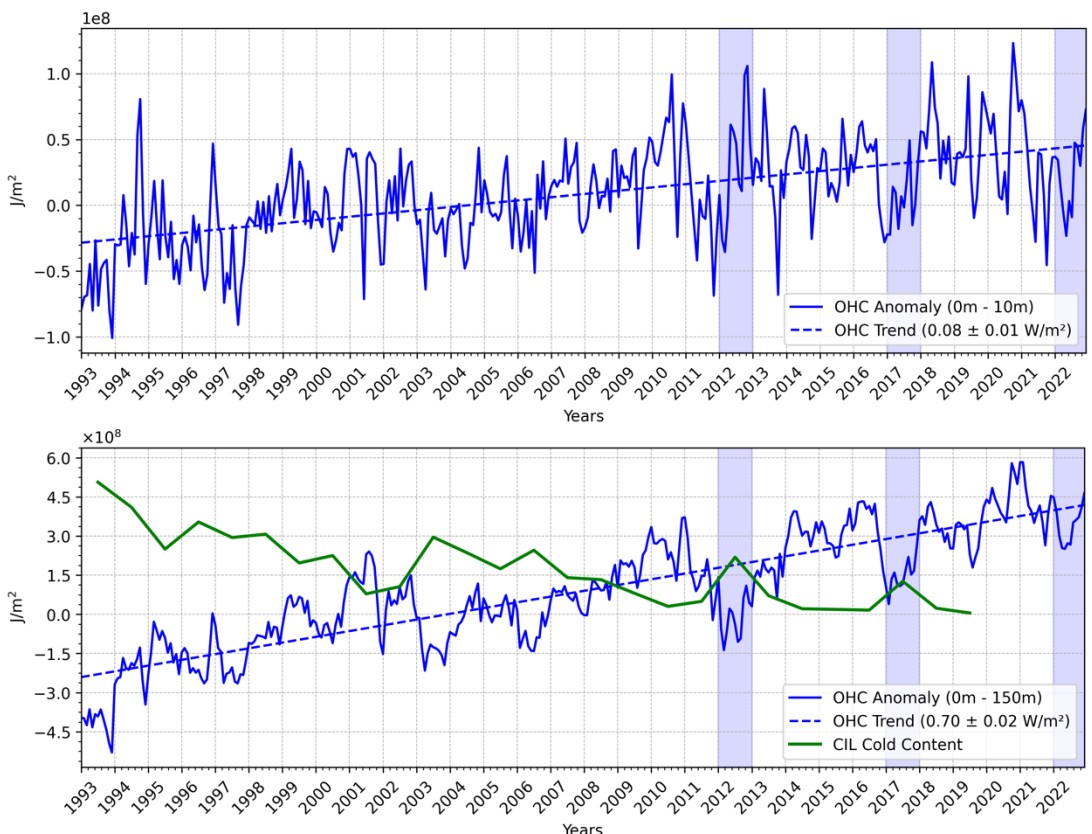

**Figure 9: Monthly basin-averaged of the ocean heat content anomalies (in J m⁻²) estimated for the BLK-REA in 0-10 m (top) and 0-100 m (bottom). The monthly ocean heat content anomalies are defined as the deviation from the climatological ocean heat content mean (1993–2014) of each corresponding month. Mean trend values are also reported for each layer (bottom right corner). In 0-100 m (bottom), the green curve corresponds to the CIL cold content from Capet et al. (2020). The blue shades highlight the recent years when the CIL is present: 2012, 2017 and 2022.**

**Table 1: Trends estimations together with the 95% confidence interval (in brackets) for the ocean heat content anomaly (W m⁻²) from BLK-REA for the periods 1993–2022 and 1993–2018, and from the previous Black Sea reanalysis (Lima et al., 2021) for the period 1993–2018.**

| | 1993-2022 | 1993-2018 | |
| --- | --- | --- | --- |
| | BLK-REA | BLK-REA | Lima et al. (2021) |
| 0-10 m | 0.08 (0.01) | 0.09 (0.01) | 0.11 (0.01) |
| 0-50 m | 0.35 (0.02) | 0.37 (0.02) | 0.45 (0.04) |
| 0-200 m | 0.74 (0.02) | 0.72 (0.03) | 0.81 (0.05) |
| 0-1000 m | 0.84 (0.02) | 0.83 (0.03) | 0.83 (0.04) |



### 3.2.2 The Black Sea overturning circulation

We follow the methodology of Ilicak et al. (2022), computing the meridional overturning circulation (MOC) in density space to better represent water mass transport in the Black Sea. We divide the water mass structure of the Black Sea in 50 different sigma2 ($\sigma_2$; potential density anomaly with [kg m⁻³] respect to a reference pressure of 2000 dbar) density bins and compute the MOC using the formula:

$$\psi^*(y, \bar{\sigma}) = -\frac{1}{T} \int_{t_0}^{t_1} \int_{x_{B1}}^{x_{B2}} \int_{-H}^{0} \mathcal{H}[\bar{\sigma} - \sigma(x, y, z, t)] \times v(x, y, z, t) dz dx dt \tag{4}$$

where $\mathcal{H}$ is the Heaviside function and $v$ is the meridional velocity. We used 100 $\sigma_2$ density bins to remap the mass flux fields.

A very narrow cell with positive values (clockwise circulation) of approximately 0.03–0.04 Sv is observed just below 26 kg m⁻³ around 43°N (Figure 10). However, at densities higher than 26 kg m⁻³, the MOC structure predominantly exhibits negative values (indicative of anticlockwise circulation), exceeding -0.03 Sv. At a density of approximately 25.5 kg m⁻³, the

MOC forms a dipole pattern, with slightly positive values between 42°N and 44.5°N and negative values south of 42°N. Above this, the circulation remains anticlockwise until approximately 23.75 kg m⁻³, where a clockwise pattern re-emerges between 23.75 kg m⁻³ and 22 kg m⁻³, with positive transport exceeding 0.02 Sv. Nonetheless, within this layer, localized negative values are observed, particularly around 45°N. In general, below 23.75 kg m⁻³, there is a predominance of anticlockwise circulation, especially in the southern part of the basin, likely associated with the inflow of Mediterranean

Water into the Black Sea. Above this isopycnal, positive values indicate a clockwise circulation, which is linked to the formation of the CIL. These findings are consistent with the results of Ilicak et al. (2022).

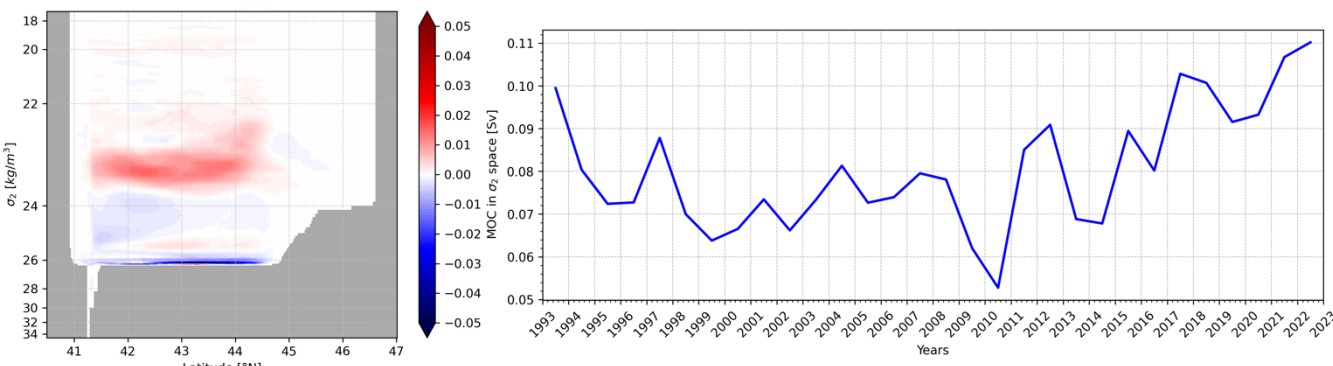

**Figure 10: Time-mean overturning transport in density space (left); Time evolution of the maximum BLK-MOC in density space between 22.45 and 23.85 kg m⁻³ (right).**

Next, we identify the maximum MOC in density space for the Black Sea between 22.45 and 23.85 kg m⁻³, corresponding to depths of approximately 25 to 80 m (Ilicak et al., 2022). The MOC declined from 0.1 Sv in 1993 to a minimum of nearly





0.01 Sv in 2010. After 2010, the MOC exhibited alternating periods of increase and decrease, but with an overall upward trend, reaching its highest values of almost 0.12 Sv in 2022. Stanev et al. (2019) reported that the Black Sea MOC has weakened over the past 30 years, possibly due to anthropogenic global warming. In recent years, the CIL has nearly

disappeared, as shown by observational data and reanalysis results (Stanev et al., 2019; Lima et al., 2021; Capet et al., 2020), as also discussed in the OHC section (Figures 8 and 9). Ilicak et al. (2022) associated the decline in MOC with the loss of CIL cold content between 1993 and 2010. However, since 2010, the MOC has started to increase, while the CIL is only present in 2012 and remains very weak in 2017 and 2022. This suggests that factors other than the CIL may be influencing the Black Sea MOC and should be investigated in detail.

**3.2.3 The Rim Current interannual variability**

The Rim Current is the dominant cyclonic gyre that defines the general circulation of the Black Sea. The Black Sea Rim Current Intensity Index (BSRCI) measures the strength of this current in a given year relative to the multi-year average. It is based on the average surface current velocity in the Rim area confined by the isobaths of 200 and 1800 m. Two sections are chosen as representative for the Rim current: a northern section between 33°E–39°E, and a southern section between

31.5°E–35°E. For each section, the BSRCI is defined as:

$$BSRCI = \frac{\overline{V}_{ann} - \overline{V}_{cl}}{\overline{V}_{cl}} \tag{5}$$

with $\overline{V}_{ann}$ the annual average surface current speed in the respective area and $\overline{V}_{cl}$ the long-term average over the period 1993-2022. In this way, the index is close to zero when the annual mean state is near normal, while positive values indicate a stronger Rim Current, and negative values represent a weaker one (Peneva et al., 2021; von Schuckmann et al., 2021). The

BSRCI OMI provides the intensity of the Rim Current in both the Northern and Southern Black Sea. In this study, we present updated results based on the latest data from the Black Sea reanalysis. The values are predominantly negative before 2010, with a notable negative peak below -0.2 in 1997 (Figure 11). After 2010, the values alternate between negative and positive, with positive peaks observed in 2014, 2017, 2020, and 2022. The BSRCI peak in 2014 exceeds 0.15 and reaches a maximum of over 0.25 in the southern branch. The intensity in both branches generally coincides, though the southern

branch typically exhibits higher values. Peneva et al. (2021) also identified a peak in 2014 using results from the previous Black Sea reanalysis. The current analysis reveals a trend of +2.9% per decade, which is a value compatible with Peneva et al. (2021) as well.

We also present the wind stress curl (Figure 11), diagnosed using the NEMO bulk formulation at the model's native resolution, based on ERA5 wind data. As previously suggested by Stanev et al. (2000) and further examined by Peneva et al.

(2021), the wind stress curl plays a key role in modulating the Rim Current. Our results support this relationship, showing strong agreement between years of enhanced mean wind stress curl and increased intensity of the Black Sea Rim Current Index (BSRCI), particularly in 2014 and 2018. This correspondence is especially evident in the southern section of the basin.



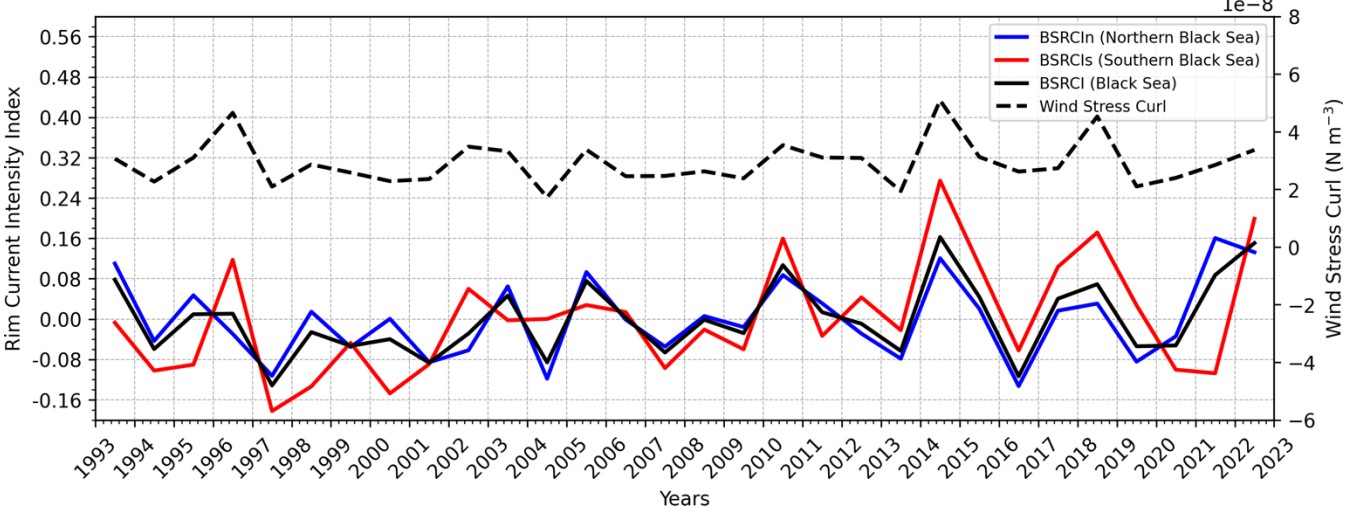

**Figure 11: Time series of the Black Sea Rim Current Index (BSRCI; black) at the north section (BSRCIn; blue), south section
(BSRCIs; red), the average (BSRCI) and its tendency for the period 1993-2020. The black dashed curve represents the annual
mean wind stress curl (N m$^{-3}$) averaged for the Black Sea based on the ERA5 reanalysis.**

## 4 Conclusions

The new BLK-REA features a higher model resolution, providing a more consistent and accurate representation of Black Sea
physics. The updated configurations include the use of LOBCs, allowing improved water exchange through the Bosphorus
Strait. This enabled further refinements in the freshwater balance to be implemented, such as incorporating hourly
precipitation data and monthly runoff for the Danube River. These improvements were not possible in the previous Black
Sea reanalysis, as its closed boundaries required a controlled freshwater balance to prevent drifts in SSH.

Overall, the BLK-REA results are highly satisfactory for key ocean variables, including T, S, SLA. T accuracy exhibits
strong seasonality, with the basin-averaged RMSD of SST reaching its lowest value in spring (0.43 °C) and its highest in fall
(0.61 °C). At depths within the seasonal thermocline, BLK-REA shows high RMSD errors for T in fall and summer, while
errors are lower in winter and spring. In contrast, salinity shows less seasonal variation, with the highest errors consistently
occurring in the 50–200 m layer throughout most of the period (Figure 2; bottom right). Occasionally, salinity errors are also
elevated at the surface. SLA errors do not exhibit clear seasonality and have remained around 0.02 m since 2000; an
acceptable level considering that the SLA observation error used in data assimilation is approximately 0.04 m. Across all
seasons, the highest SLA errors are observed along the Rim Current, primarily due to high mesoscale activity along its
pathway.

The reanalysis has proven to be an important tool for investigating the warming trend in the Black Sea, highlighting the
recent disappearance of the CIL. Both T and OHC exhibit a warming signal. In the 0–100 m layer, the warming trend is
occasionally interrupted, with decreases in OHC coinciding with periods of CIL presence, as observed in 2012, 2017 and
2022 (Figure 9). Between 1993 and 2010, the decline in CIL formation may have influenced the MOC in the Black Sea.



However, further investigations are needed to understand the recent increase in MOC and its relation with the CIL formation. Our results also reveal the significant influence of wind stress curl on the interannual variability of the Rim Current, with a particularly strong signal observed in its southern branch.

The BLK-REA dataset presented in this manuscript has been available online in the Copernicus Marine Service catalog since
December 2024 and is extended monthly in Interim mode, which applies less refined configurations for preliminary processing. The Interim results are replaced annually with an extension of the reanalysis produced using optimal configurations and the assimilation of reprocessed data, which is considered the highest quality of observations. In alignment with the Black Sea near-real-time analysis and forecasting system, preparations for the next BLK-REA are already underway. Planned improvements include the integration of the Azov Sea in the model domain and the inclusion of runoff
data from the European Flood Awareness System (Thielen et al., 2009). The plan also includes extending the reanalysis to cover previous decades, starting from 1980. This will allow for the extension of the existing OMIs and the preparation of new ones. In fact, tracking the warming signal in the Black Sea is essential, and our plan is to expand the analysis of the impacts of this warming, including monitoring marine heatwaves.

However, a major challenge is the limited availability of observations from 1980 onward. To address this gap, it may be
necessary to integrate additional in-situ datasets beyond those available from SeaDataNet and Copernicus. Therefore, continuous monitoring of the Black Sea – particularly by enhancing observation systems – is crucial for maintaining the quality of reanalysis. In recent years, ongoing advancements in observation technologies and data integration have become increasingly important to further improve reanalysis accuracy and support long-term environmental studies.

## 5 Data Availability

The BLK-REA dataset presented in this study can be found in online repositories. The names of the repository/repositories and accession number(s) can be found below: https://doi.org/10.48670/mds-00356.

## 6 Author contribution

LL led the study, built the reanalysis system, and was involved in all parts of the work. DA and MI contributed to the development of the hydrodynamical model. They also shared useful ideas that improved the study in many ways. EJ helped
set up the data assimilation strategies and also gave important suggestions to improve the work. FC helped with the validation of the reanalysis results. AS gave helpful comments that guided the research. PM supported the preparation of the in-situ observation data for the data assimilation and helped with the validation step. EC also gave useful suggestions that helped improve the work. All authors contributed to writing the article and approved the final version.



## 7 Competing interests

The authors declare that they have no conflict of interest.

## 8 Financial support

This research was funded by the Copernicus Marine Service for the Black Sea Monitoring and Forecasting Centre (Contract No. 21002L4-COP-MFC-BS-5400).

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
