# Peer review of "Advances in Monitoring Black Sea Dynamics: A New Multidecadal High-Resolution Reanalysis"

_EGUsphere, 2025_

## Author Comment (AC1)

Dear Reviewer #1, thank you for reading and suggesting modifications to our manuscript entitled "Advances in Monitoring Black Sea Dynamics: A New Multidecadal High-Resolution Reanalysis".

We believe that your review has helped to substantially improve the revised manuscript. The changes in the manuscript have been highlighted in red. Additionally, please find below a list with our point-by-point answers (*in italic*) to your comments and suggestions.

The study introduces a new high-resolution regional ocean reanalysis for the Black Sea (BLK-REA), covering 1993–2022. Key strengths include an enhanced model grid, improved boundary conditions, an updated observation-based mean dynamic topography (MDT), and refined error handling. Validation spans multiple time periods, depth levels, and subregions, and the authors effectively situate their work within the existing literature while making good use of publicly available data and tools. However, the manuscript omits any quantitative comparison with the previous reanalysis. To substantiate the claimed improvements, the authors should present figures and tables that compare error metrics from the old and new reanalyses, and from the model free-run (without assimilation but with SST nudging), using the same validation datasets and circulation diagnostics. A more fundamental concern lies in the title's promise to "monitor the dynamics" of the Black Sea: by using a sequential 3D-VAR assimilation scheme without dynamically consistent background covariances, it's unclear whether true ocean dynamical features are being captured or artificially generated. This issue could become especially critical as the authors plan to extend the reanalysis back to 1980, before satellite data were available, as trends or interannual variability could simply reflect changes in data coverage. Finally, the manuscript would benefit from careful copyediting and enhanced figure design to improve readability and visual clarity.

> *We thank the reviewer for the constructive general comments. The main goal of this work is to present the new Black Sea reanalysis (BLK-REA). Comparisons with the previous version, including detailed error metrics, are already documented in the publicly available Quality Information Document (Lima et al., 2024; [https://documentation.marine.copernicus.eu/QUID/CMEMS-BLK-QUID-007-004.pdf]). Methodologically, we adopted a 3D-Var scheme, which is still widely applied in ocean reanalyses given its robustness and computational efficiency for long integrations, while flow-dependent ensemble methods remain impractical at this resolution and timescale. The validation results included in the manuscript demonstrate that BLK-REA achieves high skill against available observations, confirming the reliability of the product for monitoring the Black Sea. Figures and captions have also been revised to improve clarity.*

Specific Comments:

Abstract and Introduction

1- The abstract and Introduction are not well structured. In both the authors discuss their previous reanalysis before describing it.

> *We appreciate the reviewer's comment. This work presents a new version of the Black Sea reanalysis developed under the Copernicus Marine Service. To improve clarity, we have made modifications to the text in both the Abstract and Introduction, enhancing readability and better highlighting the context and relevance of the new reanalysis version. We believe the revised structure supports a logical flow and helps readers*

*understand the broader scope of the work. We reviewed both sections to ensure clarity and remain open to more specific suggestions from the reviewer.*

2- The abstract highlights pronounced interannual variability in the meridional overturning circulation, but the intended message isn't clear. In particular, it's uncertain whether data assimilation has modified the overturning cell compared to the model free-run. This point is worth exploring further.

*We thank the reviewer for this observation. However, we would like to clarify that it is not uncertain whether data assimilation has modified the overturning cell. In fact, the assimilation process certainly impacts the thermohaline structure by adjusting the temperature (T) and salinity (S) fields throughout the water column. These adjustments directly affect the density field and consequently the thermohaline circulation, including the meridional overturning cell.*
*It is important to emphasize that data assimilation is intrinsic to the concept of ocean reanalysis. The model alone (i.e., a free-run simulation) is subject to significant uncertainties and biases, especially in the representation of long-term variability and vertical structure. Data assimilation aims to correct these deficiencies by continuously constraining the model with observations.*

*In our system, the state vector x includes temperature and salinity, and the assimilation of sea level anomalies (SLA) also leads to adjustments in T and S through the dynamic height operator. Therefore, the overturning circulation represented in the reanalysis reflects a dynamically consistent integration of observations and model physics, and is expected to differ significantly from a free-run.*

*Nonetheless, we have decided to remove this sentence from the abstract, considering that the results on the meridional overturning circulation are less relevant to the scope of the paper and do not need to be emphasized in the abstract.*

3- Line 47: "models that are constrained", may be "models that are driven"? also I would remove "best" from "best available observations" unless the authors had a different meaning in mind.

*We thank the reviewer for the suggestion. The text has been revised to indicate that the system is driven by atmospheric forcing and data assimilation. In other words, the model is constrained by observations to produce dynamically consistent fields that remain close to reality; a key characteristic of ocean reanalyses.*

*As for the use of "best available observations," we agree that the term could be made more precise. The reanalysis system assimilates both in situ and satellite-based observations, including multi-year (MY) or reprocessed datasets provided by the Copernicus Marine Service. These datasets are subject to rigorous quality control, bias correction, and validation, making them highly suitable for reanalysis applications.*
*To address the reviewer's concern, we have slightly revised the sentence to improve clarity and accuracy.*

*The revised sentence now reads (Lines 50-51):*

*"These products use state-of-the-art models driven by atmospheric forcing and data assimilation, which integrates high-quality multi-year satellite and in situ datasets to reconstruct historical ocean conditions."*

Methodology

1- The authors state that the current reanalysis uses monthly-averaged LOBCs from a U-TSS simulation covering 2016–2019, but the reanalysis period spans 1993–2022. Could the authors clarify how these boundary conditions are applied outside the U-TSS period, and discuss any implications for consistency over the full timeline? Have the authors tried to use more frequent LOBCs, like weekly?

*We thank the reviewer for this relevant observation. The use of monthly-averaged LOBCs from the U-TSS simulation (2016–2019) was a necessary choice due to the high computational cost of running U-TSS over the full reanalysis period. We used the 4-year U-TSS model simulation, taking advantage of the availability of boundary conditions and observational data to validate the U-TSS results during this period, rather than attempting a full 30-year simulation, which would be computationally highly demanding and impractical at the moment.*

*For this version of the reanalysis, we extended the model domain to explicitly include the exchange between the Black Sea and Marmara Sea via the Bosphorus Strait. A Marmara box was introduced into the domain, with its open boundaries prescribed by the U-TSS model. This approach enabled a two-layer dynamical response through the strait, driven by the contrasting conditions at the Black Sea and Marmara ends, with the Marmara side effectively represented by the boundary forcing. However, since U-TSS outputs are only available for 2016–2019, we applied the monthly climatology derived from this period across the entire reanalysis timeline (1993–2022), allowing for seasonal variability to be preserved at the boundary. While this does not capture interannual signals, the influence of the boundary is relatively localized, and the system's internal dynamics and data assimilation reduce long-term inconsistencies.*
*There is an ongoing effort to have more consistent LOBCs for both marginal seas connecting the Turkish Strait System. However, more frequent LOBCs (e.g., weekly) are not available at the moment, and generating them for the entire period would be currently unfeasible.*

*A clarifying note was added in the manuscript to address this limitation (Lines 138-142), as follows:*
> *"Due to computational constraints, the LOBCs at the Bosphorus Strait were derived from monthly-averaged outputs of a U-TSS simulation covering the period 2016–2019. A monthly climatology from this period was applied consistently over the full reanalysis timeline (1993–2022) to represent seasonal variability. Although this approach does not capture interannual signals at the boundary, the internal dynamics of the Black Sea, combined with the data assimilation of satellite and in situ observations, help maintain physical consistency throughout the basin.".*

2- The use of an observation-based mean dynamic topography (MDT) in the current assimilation system represents a significant methodological shift from the previous version described in Lima et al. (2021), which relied on a model-derived MDT (1993–2012). To justify

this change, the authors should briefly comment on the impact of this change, and provide supporting evidence (e.g., comparative diagnostics or sensitivity tests) illustrating how it improved SLA assimilation or the overall model skill.

*We thank the reviewer for highlighting this important methodological update. The decision to adopt an observation based mean dynamic topography (MDT) in the current reanalysis was supported by sensitivity tests comparing its performance with that of the model derived MDT used in the previous version (Lima et al., 2021). These tests showed that the observation based MDT led to slightly improved SLA assimilation, particularly in terms of reduced RMSD values and better agreement with along-track altimeter data.*

*Further comparative results between the current reanalysis and the previous version are provided in the Quality Information Document (Lima et al., 2024), available at [https://documentation.marine.copernicus.eu/QUID/CMEMS-BLK-QUID-007-004.pdf]. Specifically, Section VI and Figure VI-6 illustrate improvements in SLA skill metrics, where the current system, using the observation based MDT, shows lower RMSD errors, especially in the period after 2000.*

*We have added a brief clarification in the manuscript to justify this methodological choice (Lines 173-177).*
> *"In contrast to the previous version of the reanalysis (Lima et al., 2021), which used a model-derived mean dynamic topography (MDT), an enhancement in the present version is the use of an observation-based mean dynamic topography MDT to compute the model-equivalent SLA in data assimilation. Sensitivity tests indicated that this choice improves the assimilation skill of SLA, leading to systematically reduced RMSD values. The new MDT field is available in the Copernicus Marine Service catalog: https://doi.org/10.48670/moi-00138.".*

3- The presentation of the cost function (Equations 1 and 2) should include a description of the terms: what does x include?, the choice of the components of observational errors R, etc. What the authors mean by "use the same values for S and T" in Line 133? And what threshohld did they apply for data rejection in line 135?

*We thank the reviewer for the helpful observations. Please find our responses below:*

1. *State vector $x$: We have clarified the definition of the state vector x in the main text. The revised manuscript now includes the following sentence in Lines 186–188:*
   > *"OceanVar is a multivariate scheme, i.e., the state vector x can contain the following model state variables: temperature (T), salinity (S), sea level anomaly (SLA), and horizontal velocities (u and v). However, only the first three variables are employed in the present BLK-REA implementation."*

2. *Observational errors and "same values for T and S": While the overall methodology follows the previous version of the Black Sea reanalysis (Lima et al., 2021), we agree that further clarification is needed. The instrumental errors for in situ T and S are depth-dependent, derived from Ingleby and Huddleston (2007), and assume different values for the two variables. Representation errors, on the other hand, are implemented as a multiplicative factor applied to*

these depth-dependent errors, with spatial variation guided by prior model–observation statistics. In this component, the same spatially varying multiplicative factor is applied to both T and S. Although this is a simplification, it is often adopted when the two variables exhibit similar observational sampling patterns and spatial scales. We revised the manuscript to clarify this distinction and to briefly explain the nature of representation errors, which reflect unresolved subgrid variability and structural model differences not captured by instrumental noise alone

3. *Threshold for data rejection: The data quality control and rejection criteria are exactly as described in Lima et al. (2021). Given that no changes were made in this aspect, we did not repeat the details, but we have reinforced this reference in the manuscript.*

   *We have expanded the description of the observation error components and clarified the usage of equal representativity errors for T and S, as well as the quality control procedure. The modified section appears in Lines 155–167, and now reads:*

   > "In data assimilation, the in situ instrumental errors assume different values for T and S and vary in the vertical dimension based on statistics derived from Ingleby and Huddleston (2007). The in situ representation errors are defined as a multiplicative factor applied to the depth-dependent instrumental errors and vary horizontally on the model grid according to previous model–observation statistics. In this component, the same spatially varying factor is applied to T and S, which is a simplification justified by the similar spatial sampling patterns and statistical structure of the T/S in situ observational datasets. Representation errors account for unresolved physical processes, subgrid-scale variability, and model errors that are not part of instrumental uncertainties. Both components of in situ errors are kept constant over time. For SLA observations, the instrumental error is set to 4 cm, and the representation errors vary spatially and monthly following Oke and Sakov (2008). The quality control procedures and data rejection thresholds are applied as described in Lima et al. (2021), with no changes introduced in BLK-REA."

4- The choice of specific multi-year periods EOFs (1984–1993, 1994–2003, etc.) should be explained. Why these particular time blocks were selected? Were alternative choices considered? What about the seasonal aspects?

*We thank the reviewer for this pertinent question. The data assimilation method used in our system is based on 3DVar, which relies on a background error covariance matrix that is static and derived from past model integrations. Since this matrix does not evolve in time, using a single EOF-based covariance structure for the entire reanalysis period (1993–2022) would not adequately capture decadal shifts in variability and circulation patterns.*

*To address this, we selected decadal blocks (1984-1993, 1994-2003, etc.) as representative periods for constructing the background error covariances, aiming to*

*better reflect changes in the system's dynamics over time. These blocks were chosen to balance the need for temporal representativeness with the statistical robustness required to compute monthly EOFs. The choice of decadal updates is also supported by changes in observational availability, which varies over time and affects the accuracy of the covariance estimation, since EOFs derive from an integration with data assimilation.*

*Seasonal variability is inherently included, as for each decade we compute separate EOF sets for each month (January to December), as described in the manuscript (Lines 208–209). This ensures the assimilation system respects the seasonal structure of background errors.*

*Some sensitivity tests were conducted using background error covariance matrices derived from different model integrations, both with and without data assimilation. Although, for our approach, we consider that using an integration with assimilation, in which the numerical representation is corrected by observational constraints, provides a more realistic estimate of background errors, the experiments showed minimal differences in assimilation performance when using covariances derived from either type of integration. This outcome further supports the robustness of our chosen method.*

5- The estimation of representation errors for SLA and T/S observations is not fully detailed. Since salinity errors are larger in the upper 200 m despite LSBC being applied below 700 m, could you clarify how these errors are specified and whether dynamic bias correction or adjustments to the prior covariance could further reduce the persistent salinity biases?

*It is important to note that these representation errors are not intended to correct systematic model biases. As previously discussed in our answer to question 3 above, representation errors for T, S, and SLA account for unresolved processes, subgrid-scale variability, and model limitations. They are derived from observational statistics while taking into account the model's spatial resolution (see Lines 155–167 in the revised manuscript).*

*Nevertheless, all data assimilation systems are affected by systematic biases arising from imperfect numerical models, sparse observations, and inherent limitations of the assimilation scheme. In periods with very sparse in situ observations, these biases can generate artificial T/S drifts over the course of the integration.*

*To avoid reanalysis T/S drifts in deeper layers where observational coverage is very sparse, we applied the Large Scale Bias Correction (LSBC) below 700 m, which is an independent scheme and not part of the data assimilation. Above 700 m, the reanalysis is left free from LSBC constraints because applying them in the upper layers could artificially degrade natural temperature and salinity trends. In these upper layers, we rely on the assimilation system, which integrates observations with strict quality control to correct model errors and improve the overall representation of the model. Assimilation corrections are localized and observational coverage is sparse, so persistent salinity biases in the upper 200 m are likely influenced by uncertainties in the freshwater budget, including simplified river runoff and precipitation forcing intrinsic to the model, rather than by mis-specified representation errors within the assimilation system.*

Results

1- The authorsthe analysis error for validating the reanalysis. I would suggest using the forecast error, or at least discussing it.

> *We appreciate the reviewer's suggestion. However, we would like to clarify that this study presents a reanalysis product rather than a forecasting system. In this context, the evaluation is based on daily mean reanalysis fields over the period 1993 to 2022, which are the most representative outputs for validating a reanalysis. The use of forecast error is typically more applicable to forecast skill assessments. In contrast, reanalyses incorporate observations directly into the model trajectory. Their fields, especially daily mean values, are widely used in validation because they represent the best estimate of the ocean state at each time step.*

2- In Section 3.1 and Figure 2 (top), you show that temperature RMSD exceeds $2\,°C$ in the upper $50\,m$ during summer, suggesting a persistent misrepresentation of the seasonal thermocline. Could you provide some discussion of possible causes and comment on whether any sensitivity tests were explored?

> *We acknowledge the reviewer's observation regarding the elevated RMSD values in the upper $50\,m$ during summer and agree this reflects a persistent challenge in representing the seasonal thermocline. This misrepresentation primarily originates from the model component of the reanalysis system. It is a well-known issue in ocean modeling, where difficulties in accurately resolving vertical stratification and thermocline sharpness are common across various models and regional applications. Some physical parameterizations, such as vertical mixing schemes, surface heat flux formulations and even atmospheric forcing fields may contribute to these discrepancies and are a subject of ongoing improvement in the Black Sea reanalysis development. In addition to the mixed layer parameterization, the representation of mesoscale eddies also contributes to surface intensified RMSD. Spatial representation of the mesoscale eddies is an ongoing research field in ocean modeling. Although this remains an area for refinement, we note that the temperature errors in this version have been reduced compared to the previous reanalysis, including within the thermocline region. This improvement is documented in the Quality Information Document (Lima et al., 2024), particularly in Section II, Figure VI-2:*
> *https://documentation.marine.copernicus.eu/QUID/CMEMS-BLK-QUID-007-004.pdf*
>
> *No dedicated sensitivity tests were performed in this version to isolate the cause of the thermocline bias, but addressing it remains a key consideration for future system updates. For instance, we are currently testing a new bulk formulation for atmospheric forcing to reduce biases in the northern coast of Turkey where significant upwelling-related biases have been observed, as discussed in the next comment.*

2- In Section 3.1, the authors note that the intensified upwelling and resulting SST biases along the Anatolian coast may stem from the bulk formulation, and that refinements are planned in future versions. It would be helpful if you could briefly elaborate on what aspects of the formulation you intend to refine or test, and whether any preliminary sensitivity experiments have been conducted to assess their impact.

*The current MFS bulk formulation uses a drag coefficient computed according to Hellerman and Rosenstein (1983) for wind stress components. The drag coefficient can be simply computed as*

$$C_D = \alpha_1 + \alpha_2(u^2 + v^2)^{1/2} + \alpha_3(T_a - T_s) + \alpha_4(u^2 + v^2) + \alpha_5(T_a - T_s)^2 + \alpha_6(u^2 + v^2)^{1/2}(T_a - T_s)$$

*where $\Delta T = T_a - T_s$ is the difference between air temperature ($T_a$) and sea surface temperature ($T_s$). The values of $\alpha$ are fitted as described in Rosenstein (1983). Under conditions that favor upwelling, a positive feedback loop can occur. When the sea surface temperature cools, the difference between the air temperature and the surface temperature ($T_a - T_s$) increases. This larger temperature difference causes the drag coefficient to increase, which in turn strengthens the upwelling-favorable winds. Stronger winds lead to further cooling of the SST, causing the temperature difference to become even greater and perpetuating the cycle. However, other bulk formulas such as NCAR or ECMWF in NEMO use $C_D$ formulation, only a function of wind velocities (u and v) in which case this positive feedback does not exist. We are currently testing the NCAR drag coefficient formulation in the MFS bulk formulas.*

3- Figure 2 (bottom) and Figure 4 highlight that salinity RMSD can exceed 1 psu near the surface, particularly in the western Black Sea region. Could you elaborate on whether these discrepancies may be attributed to biases or uncertainties in freshwater forcing? Done!

*We thank the reviewer for this observation. The elevated surface salinity RMSD in the western Black Sea is likely influenced by uncertainties in the freshwater budget. Precipitation is derived from the ERA5 atmospheric reanalysis, which may contain regional biases, while evaporation is computed online via the model's bulk formulae. River discharge is prescribed using monthly climatologies for most rivers, with the Danube being the only exception where monthly-varying values are used. These limitations in the atmospheric and riverine forcing, along with model-intrinsic processes (e.g., vertical mixing), can contribute to surface salinity discrepancies, especially in regions strongly influenced by freshwater inputs, like the Black Sea.*

*We added the following sentence in Lines 268–271 of the revised manuscript:*
> *"Unlike temperature, the Hovmöller diagram of RMSD for salinity does not exhibit a clear seasonal cycle. Errors exceed 1 psu during short periods (Figure 3), particularly in the upper layers, where high uncertainties are likely derived from precipitation biases in ERA5, simplified river runoff forcing, and the internal freshwater budget computed via the model's bulk formulae."*

4- In Section 3.1 and Figure 7, the authors note that reanalysis-derived EKE values are substantially higher than those derived from altimetry, especially in the Batumi and Sevastopol eddies. I cannot follow the authors reasoning that this is due to the coarse resolution of the data. This is more likely to be coming from the model being over-energetic in resolving mesoscale variability, which in this case needs to be better discussed and analyzed in the manuscript. Done!

*We thank the reviewer for this observation. We confirm that EKE was computed from the geostrophic component derived from reanalysis SSH, ensuring a clean and consistent comparison with the EKE calculated from the Level-4 altimetry product.*

*While the coarser 0.25° resolution of the L4 fields may reduce mesoscale variance, this is not the only factor explaining the difference. Model dynamics can also lead to higher EKE values, particularly if mesoscale features are more energetic than in observations. Such behavior is not uncommon in high-resolution ocean models, especially in regions with intense eddy activity such as the Batumi and Sevastopol eddies. Both factors likely contribute to the larger EKE amplitude in BLK-REA.*

*We have revised the text in Section 3.1 (Lines 363–372) accordingly:*
> *This interpretation is further supported by EKE maps (Figure 8), which consistently show high energy levels in these regions, indicating intense mesoscale activity that aligns with the SLA discrepancies viewed in Figure 7. EKE maps show elevated values along the Rim Current, particularly along the Caucasian coast. To ensure a clean comparison with altimetry-derived estimates, BLK-REA EKE was computed from the geostrophic component only, using SSH from the model. BLK-REA EKE values generally exceed those derived from altimetry, especially in the Batumi and Sevastopol eddies, where peak values reach about 300 cm² s⁻² in fall and spring, respectively, compared to altimetry-based estimates of around 220 cm² s⁻². This difference likely reflects multiple factors, including the spatial smoothing inherent to the 0.25° Level-4 altimetry product, the reduced capability of altimetry to capture smaller-scale variability due to mapping and interpolation procedures, and possible over-energetic behavior of the model arising from its higher resolution, physical parameterizations, and potentially insufficient dissipation of mesoscale energy.*

5- It would be useful to check the quality of the figures and captions. For instance, the font in Figure is too small. The caption of Figure is not clear. The title of Figure 7 "for layer: surface"?

> *We thank the reviewer for this comment. We have carefully checked all figures and captions in the revised manuscript. Figure fonts have been increased for better readability, captions have been clarified, and Figure 8 has been corrected to properly indicate that the results correspond to the surface.*

- Sections 2.2 and 3.1: To help readers better understand data coverage and its impact on validation, it would be useful to indicate how many in-situ profiles are typically assimilated per cycle and to quantify observation density over time (e.g., profiles per year or season).

> *We thank the reviewer for this suggestion. In the revised manuscript, Section 2.4 now includes a new Figure 1 showing SST increments together with the number of T/S profiles and SLA along-track observations within a single assimilation cycle. Furthermore, Section 3.1, in the validation figures, includes Hovmöller diagrams showing the temporal evolution of observations and spatial maps indicating the number of observations by region and season. These additions provide a clearer view of data coverage and its impact on validation.*

**References**

Hellerman, S., and M. Rosenstein, 1983: Normal monthly wind stress over the world ocean with error estimates. *J. Phys. Oceanogr.*, 13, 1093-1104.

Lima, L., Ciliberti, S. A., Aydoğdu, A., Masina, S., Escudier, R., Cipollone, A., Azevedo, D., Causio, S.; Peneva, E., Lecci, R.; Clementi, E., Jansen, E., Ilicak, M.; Cretì S., Stefanizzi, L., Palermo, F., & Coppini, G. (2021) Climate Signals in the Black Sea From a Multidecadal Eddy-Resolving Reanalysis. Front. Mar. Sci. 8:710973. doi: 10.3389/fmars.2021.710973.

Lima, L., Azevedo, D., Ilicak, M., Jansen, E., Costa, F., Causio, S., Cretí, S., and Clementi, E.: EU Copernicus Marine Service Quality Information Document for the Black Sea Physics Reanalysis, BLKSEA_MULTIYEAR_PHY_007_004, issue 5.0, Mercator Ocean International, https://documentation.marine.copernicus.eu/QUID/CMEMS-BLK-QUID-007-004.pdf, last access: 28 August 2025, 2024.

---

## Author Comment (AC2)

Dear Reviewer #2, thank you for reading and suggesting modifications to our manuscript entitled "Advances in Monitoring Black Sea Dynamics: A New Multidecadal High-Resolution Reanalysis".

We believe that your review has helped to substantially improve the revised manuscript. The changes in the manuscript have been highlighted in red. Additionally, please find below a list with our point-by-point answers (*in italic*) to your comments and suggestions.

This manuscript presents a new high-resolution multidecadal reanalysis for the Black Sea (BLK-REA), incorporating key advancements in model resolution, lateral boundary conditions, and data assimilation techniques. The study delivers an extensive validation of the reanalysis and discusses its application to ocean monitoring indicators (OMIs) such as ocean heat content, the Cold Intermediate Layer (CIL), Rim Current variability, and meridional overturning circulation (MOC). These contributions are highly relevant and timely, particularly in the context of climate-related changes in regional semi-enclosed basins.

However, despite its technical strengths, the manuscript requires significant revision before it can be considered for publication. Several methodological and interpretative aspects need clarification, refinement, or deeper discussion. I recommend Major Revisions to address the concerns outlined in the detailed comments. My detailed comments are followed below:

1. The manuscript states in line 178 that a "quasi-independent validation" was performed. However, for an objective and rigorous validation, the use of fully independent observational datasets is typically recommended. In particular, the use of observations that were excluded from data assimilation due to background quality control raises concern about possible contamination by erroneous data. For instance, in Figure 2, there are conspicuous, vertically extensive anomalies in temperature bias around 1998 and salinity bias in 1996, which could be indicative of problematic observational data rather than model performance. A discussion on the possibility of observation-induced artifacts and how such risks were mitigated is strongly recommended.

    *We thank the reviewer for this valuable comment and fully agree that, for an objective and rigorous validation, the use of fully independent observational datasets is typically recommended. In our study, we used the term quasi-independent validation to indicate that the validation includes both observations assimilated in the system, which does not constitute a fully independent validation, but also those excluded by the background quality control, thereby encompassing all available observations. We note that all datasets, whether assimilated or used only in validation, had already passed standard quality control procedures prior to assimilation. Nevertheless, some high errors observed in the T/S Hovmöller during the validation stage (Figure 3), around 1996 for salinity and 1998 for temperature, may be associated with observations that were not assimilated but are included in the validation dataset. These errors are likely due to the presence of such data, but also to sparse observational coverage, and local model biases. These points have been clarified in the revised manuscript.*

    *Revised manuscript text (Lines 162–167):*
        *"Similar to the previous version, a background quality check is implemented in the data assimilation system to reject observations that deviate significantly from the model prior solution. Rejection by*

> *background quality control does not necessarily indicate erroneous data, but often reflects large innovations that would otherwise introduce undesirable shocks in the model state. The quality control procedures and data rejection thresholds are applied as described in Lima et al. (2021), with no changes introduced in BLK-REA."*

*and Lines 280-282:*

> *During the validation stage, the elevated T/S errors around 1996 (salinity) and 1998 (temperature) in the Hovmöller diagram (Figure 3) may be linked to observations that were not assimilated but included in the quasi-independent validation. These peaks are likely due to such observations, but also sparse coverage, and local model biases.*

2. In Figures 3 and 4, temperature and salinity biases in the Eastern region show a sudden increase below 700 m, which is unexpected. Generally, variability and errors tend to decrease with depth. The authors should examine whether these anomalies arise from insufficient data qulity control, issues in model initialization, or perhaps unresolved physical processes. An explanation or hypothesis for these unusual patterns would improve credibility.

> *We note the unexpected increase in temperature and salinity biases below 700 m in the Eastern region. This behavior likely results from a combination of factors: sparse observational coverage in deeper layers before 2003 (as already shown in the Hovmöller diagrams, Figure 3), limitations in model initialization, and unresolved physical processes. Even with strict quality control, some measurement errors or inconsistencies may have persisted in the assimilated data. The introduction of Argo profiling floats from 2003 improved deep-layer observations, revealing biases that were previously undetected and not corrected. Additionally, some observations excluded from assimilation were still included in validation, potentially contributing to the apparent increase in errors.*
>
> *Revised manuscript addition (Lines 326-332):*
>
> > *"In the Eastern region, a slight increase in temperature and salinity biases occurs below 700 m (Figures 4 and 5). This pattern likely arises from sparse deep-layer observations prior to 2003 (as shown in the Hovmöller diagrams, Figure 3), limitations in model initialization, and unresolved physical processes. The introduction of Argo profiling floats from 2003 increased deep data coverage, revealing biases that were previously undetected and never corrected; below 700 m, LSBC toward WOA2018 climatology is not sufficient to constrain the model. In addition, some observations excluded from assimilation were still included in validation, further contributing to the apparent increase in errors. Even with strict quality control, some measurement errors or inconsistencies may have persisted.*

3. Although the Eastern region has the highest number of salinity observations (Figure 4, right), the RMSD and bias near the surface are larger than in other regions. The manuscript does not address this, and a discussion is warranted. Potential causes could include persistent local biases in atmospheric forcing, river discharge representation, or misrepresentation of mixing processes.

*We thank the reviewer for this comment. Although the comment refers to the Eastern region, the elevated near-surface salinity RMSD and bias actually occur in the Western region (Figure 5). These errors are mainly due to limitations in freshwater inputs from major rivers: only the Danube runoff is represented with monthly varying discharge, while other rivers follow climatologies without intra-annual variability, causing persistent local biases. Also, complex hydrodynamics near the Bosphorus Strait are included through prescribed boundary conditions, but uncertainties in these inputs can further increase local errors. Data assimilation can partially improve the model representation under these circumstances by incorporating observations with strict quality control..*

*Revised manuscript addition (Lines 319-325):*
> *"The same analysis for salinity is shown in Figure 5. The Western region exhibits the highest near-surface RMSD, with values reaching over 0.8 psu in the upper 100 m, despite high observational coverage. In contrast, RMSD values are lower in the central and Eastern regions, remaining below 0.2 psu in the upper layers. These elevated values in the Western region are mainly due to limitations in freshwater inputs from major rivers: only the Danube uses monthly varying discharge, while other rivers follow climatologies without intra-annual variability, producing persistent local biases. Prescribed boundary conditions near the Bosphorus Strait improve the physical representation but also introduce uncertainties. Data assimilation can partially improve model representation under these circumstances by incorporating observations with strict quality control, but some errors can still persist".*

4. The sentence "Bias values are in general small, and in particular in the western region" seems inconsistent with the plotted salinity bias in Figure 4, where large positive biases are evident near the surface in the western region. This discrepancy should be corrected or clarified.

> *We thank the reviewer for this observation and we agree that the original wording was inconsistent with the plotted salinity bias in Figure 5. In the revised version, we have corrected the text to better reflect the results, emphasizing that salinity biases are large in the western region near the surface (Lines 302-312).*

5. The description of how SLA is assimilated (e.g., through the dynamic height operator and use of MDT) could be elaborated further. It is unclear how the model handles vertical mapping and whether SLA assimilation is constrained only over deep waters. Furthermore, in Figure 5, the RMSD of SLA decreases after 2000-likely due to increased availability of satellite altimetry data. This hypothesis should be explicitly stated and supported with quantitative or qualitative evidence. The temporary increase in SLA RMSD around 2016 is notable but unexplained. Likewise, elevated RMSD near the coastline (Figure 6) may stem from the fact that SLA assimilation is performed only in areas deeper than 1000 m, thus leaving high-variability coastal regions unconstrained. A discussion of these spatial and temporal variations in SLA errors is necessary to clarify the model's performance boundaries.

*We thank the reviewer for these very valuable comments regarding the assimilation and validation of SLA. We have expanded the description of the SLA assimilation methodology and clarified the interpretation of the SLA validation results, following the reviewer's suggestions:*

*In the revised manuscript, we now explain that SLA increments are applied through a balance model (dynamic height operator) that imposes local hydrostatic and geostrophic balance among SLA, temperature, and salinity increments. A "level of no motion" is assumed at 1000 m depth, below which horizontal velocities are considered negligible. Following Storto et al. (2011), we further specified that SLA assimilation is performed via a local hydrostatic adjustment scheme: SLA increments, proportional to the vertically integrated density increments, are partitioned into thermo- and halosteric contributions and then distributed along vertical T/S profiles according to the background-error vertical covariances, thereby restricting SLA assimilation to deep-water regions where the balance assumptions are valid.*

*We thank the reviewer for this observation. We agree that the decrease in SLA RMSD after 2000 is closely linked to the increased availability of satellite altimetry data, as indicated by the rise in the number of observations (Figure 6). At that time, in situ T/S profiles were still scarce (see number of T/S observations in Figure 3), so the model constraints were essentially driven by SLA assimilation. On the other hand, the temporary increase in SLA RMSD around 2016 coincides with the larger availability of Argo profiles (see number of T/S observations in Figure 3). Their assimilation, in combination with SLA, appears to have degraded the SLA skill, contributing to the observed increase in RMSD. This effect likely reflects the multivariate nature of the system. Nevertheless, the SLA RMSD remained below the instrumental error level of 0.04 m used in the assimilation. These points have been clarified in the revised manuscript.*

*The revised text now contains sentences to detail the SLA assimilation, which follows Lima et al. (2021) and Storto et al. (2011 (Lines 200-203):*

> *The dynamic height operator in $V_\eta$ imposes local hydrostatic and geostrophic balance among SLA, temperature, and salinity increments, following Storto et al. (2011), with a level of no motion assumed at 1000 m, where this balance is valid. This restricts SLA assimilation to deep-water regions.*

*and in Lines 333–339, we have refined the description of SLA validation and model skill:*

> *The time series of spatially averaged SLA RMSD (Figure 6) shows strong skill after 2000, with errors around 0.02 m, below the instrumental error of 0.04 m used in assimilation. This improvement reflects the increased availability of satellite altimetry, while in situ T/S profiles remained scarce, leaving SLA as the main constraint on the model. Instead, the slight RMSD increase around 2016 coincides with the larger availability of Argo profiles. Their assimilation, together with SLA, may have slightly degraded SLA skill due to the multivariate nature of the system. Nevertheless, the RMSD remains well within acceptable limits, and these points are intended to provide an overview of SLA*

*performance rather than a detailed attribution of small temporal fluctuations.*

6. The recovery of the Black Sea Meridional Overturning Circulation (MOC) after 2010 is intriguing, especially given the concurrent decline or near-disappearance of the Cold Intermediate Layer (CIL). Although the authors note this decoupling, a deeper discussion on alternative dynamics that might drive the MOC increase (e.g., changes in wind-driven circulation, lateral advection, or water mass transformation) would be beneficial. The potential use of age-tracer or Lagrangian particle tracking experiments to explore such processes could be suggested for future work.

> *We thank the reviewer for the suggestion. An ideal age tracer release would be a huge asset, not just for understanding the Meridional Overturning Circulation (MOC) but also for revealing the Black Sea's ventilation processes and how its deep basin changes over time. Additionally, we could use spiciness to investigate if the temperature increase is being compensated for by the increase in salinity which might be due to E-P budget changes. We also add the following part in the discussion (Lines 466-472):*
>
> > *"Different water mass transformations could be the potential mechanisms behind an increase in the MOC. Specifically, an increase in salinity could compensate for the decrease in the formation of cold, dense water, which would otherwise weaken the circulation. Reanalysis model results show that there is an upward trend of SSS in the Black Sea. In addition, running multiple cycles of decadal reanalysis simulations is likely necessary to achieve a more accurate spin-up of the deep ocean. However, to investigate the detailed dynamics of the meridional overturning circulation (MOC) in the Black Sea, further research is required. This should involve using multi-cycle reanalysis model simulations combined with passive tracers, such as ideal age, to better understand the circulation patterns and timescales."*

7. Much of the text after line 419 in the "Conclusions" section reads more like future planning than summarizing key findings. It may be clearer and more structured to rename this section "Discussion and Outlook", where the manuscript first summarizes the conclusions, then outlines future improvements and data needs.

> *We have renamed Section 4 to "Discussion and Outlook" and revised the text in Lines 103–104 in accordance with the reviewer's suggestion.*

**References**

Lima, L., Ciliberti, S. A., Aydoğdu, A., Masina, S., Escudier, R., Cipollone, A., Azevedo, D., Causio, S.; Peneva, E., Lecci, R.; Clementi, E., Jansen, E., Ilicak, M.; Cretì S., Stefanizzi, L., Palermo, F., & Coppini, G. (2021) Climate Signals in the Black Sea From a Multidecadal Eddy-Resolving Reanalysis. Front. Mar. Sci. 8:710973. doi: 10.3389/fmars.2021.710973.

Storto, A., Dobricic, S., Masina, S., and Di Pietro, P. (2011). Assimilating along-track altimetric observations through local hydrostatic adjustment in a global ocean variational assimilation system. Mon. Weather Rev. 139, 738–754. doi: 10.1175/2010mwr3350.1